# Bio-inspired counter-current multiplier for enrichment of solutes

Kyle Brubaker[1], Armand Garewal [2], Rachel C. Steinhardt[1] & Aaron P. Esser-Kahn[1]

Improving the efficiency of gas separation technology is a challenge facing modern industry, since existing methods for gas separation, including hollow-fiber membrane contactors, vacuum swing adsorption, and cryogenic distillation, represents a significant portion of the world's energy consumption. Here, we report an enhancement in the release rate of carbon dioxide and oxygen of a thermal swing gas desorption unit using a counter-current amplification method inspired by fish. Differing from a conventional counter-current extraction system, counter-current amplification makes use of parallel capture fluid channels separated by a semipermeable membrane in addition to the semipermeable membrane separating the capture fluid channel and the gas release channel. The membrane separating the incoming and outgoing fluid channels allows gas that would normally exit the system to remain in the desorption unit. We demonstrate the system using both resistive heating and photothermal heating. With resistive heating, an increase in release rate of 240% was observed compared to an equivalent counter-current extraction system.

[1] Institute of Molecular Engineering, University of Chicago, Chicago, IL 60637, USA. [2] Department of Chemistry, University of California - Irvine, Irvine, CA 92697, USA. Correspondence and requests for materials should be addressed to A.P.E-K. (email: aesserkahn@uchicago.edu)

Concentration of dilute solutions is a central problem in separations ranging from protein isolation to concentration of commodity gases[1–4]. Current methods for gas concentration including cryogenic distillation, vacuum swing adsorption, and chemo-selective permeable membranes have each advanced modern industry. The various methods of gas separation have grown in use to the point where they account for 10–15% of the world's energy consumption. Lively and Scholl recently noted that more efficient methods of separation "could save 100 million tonnes of carbon dioxide emissions and US $4 billion in energy costs annually."[5]

There are several methods for gas separation, including metal and persistent organic frame-works, perm-selective membranes, and electro-chemical methods[6–9]. These methods include both separation at the molecular level and processing of gas at the micro-level. For all of these, the best method remains counter-current and cross-current exchange using hollow-fiber membrane contactors or micro-channels[2, 10]. Here, a solution high in solute concentration is separated by a semipermeable membrane from a fluid with a low solute concentration flowing in the opposite direction. For traditional counter-current extraction, the output concentration is limited to the partial pressure of the gas above a bulk phase of the carrier fluid at the release conditions used. In contrast, counter-current amplification allows the output concentration to reach many times that of bulk phase partial pressures. Used by many deep-sea fish to maintain neutral buoyancy even at great depths, counter-current amplification can drive oxygen from ambient concentrations of 0.0003 M to 3.9 M in the swim bladder[11–15]. The key to attaining these concentrations is a series of looped blood vessels between which gas can diffuse combined with a gas release trigger. Blood flowing into the swim bladder system passes by a loop adjacent to a gas storage bladder where the release of oxygen from hemoglobin is triggered due to the interaction of released lactic acid and a pH-sensitive hemoglobin. Now rich in free oxygen, the blood flows back alongside the incoming channel where free oxygen can diffuse into the incoming channel, increasing the concentration of hemoglobin-bound oxygen. The cycle repeats, successively increasing the oxygen concentration in the loop until equilibrium is reached (Fig. 1a). The counter-current amplification process produces a higher output concentration for a given release trigger than could be obtained in bulk solution.

Here, we demonstrate the possible benefits that can be had from implementing counter-current amplification compared with an equivalent counter-current-only contactor. The device shown has not been optimized and is not intended to compete directly with existing commercial implementations. We show an engineered swim bladder that employs a general mechanism of counter-current amplification to extract $CO_2$ and $O_2$. We fabricate a model counter-current amplifier consisting of a single, looped channel, separated by a gas permeable membrane (Fig. 1b, c). To trigger a change in solubility, we apply thermal or photothermal energy to one face of the amplifier (Fig. 1a). Counter-current amplification offers an alternative process to supplement current separation methods, with two unique features that build on counter-current exchangers. First, a counter-current amplification system combines the release process with a concentration step. In biological systems, this coupled release/concentration minimizes input energy and footprint. Second, the amplification can achieve very high purity through selective binding and release. In our proof-of-concept methodology, we demonstrate that counter-current amplification can be performed in a single device with gas concentrations reaching up to 1 atm from a continuously running counter-current amplifier with an internally measured temperature of 40 °C. We observe a 240% increase in gas release rate when comparing a device where gas and heat are allowed to diffuse between adjacent capture fluid channels with a device where no diffusion takes place.

## Results

**Device design.** In designing the counter-current amplifier, we pursued a general test platform to address a wide range of target solutes and solutions. In the swim bladders of physoclist fish, the amplification of solute concentration is the result of a counter-current vascular exchanger, the rete mirabile, where vessels are looped back onto themselves—separated by a thin semipermeable membrane. Release of solute from a carrier fluid (triggered by a reduction in blood pH) in the loop creates a concentration gradient between the incoming and outgoing vessels. This looped gradient drives a positive feedback where the solute concentration entering the looped (release) section increases, resulting in a larger release of solute in the looped section (release section, Fig. 1)[14]. To build an artificial system, our design requires: (i) a compatible carrier fluid; (ii) a release trigger that changes solubility; and (iii) easy fabrication for modularity. For solutes, we selected $CO_2$ and $O_2$ for their industrial and health importance, and well-studied capture solutions. Due to the photothermal/thermal trigger, our system, unlike that of the fish, uses the transfer of both heat and gas across the membrane separating the incoming and outgoing channels of the device to drive amplification (Fig. 1). As such, to understand how our synthetic counter-current amplification results in a higher gas release rate, we determine the contributions of both heat and gas to our observed amplification.

We broke the construction of the counter-current amplifier up into four main components, the counter-current exchange zone, where diffusion takes place between the incoming and outgoing fluid channels, the release zone, where the solute of interest can escape from the system, the capture fluid, and the release trigger mechanism (Fig. 1c). Combined, these features replicate the mechanism of amplification described above. These components were recreated in the design seen in Fig. 1d. Acrylic (5 mm thick) was chosen as the structural material due to the low permeability of both $CO_2$ and $O_2$, and the ease of machining small channels in the material[16]. Polydimethyl siloxane (PDMS, 140 μm thick) was chosen as the membrane material due to the ease of fabrication of varying thicknesses and the relatively high permeability of both $CO_2$ and $O_2$ (3800 and 800 Barrer, respectively)[17]. Gas selectivity in our test system was driven by the carrier fluid and the source gas, and the effect of this system on contaminant gas concentration was not investigated. The behavior of contaminants in our system likely depends on the change in solubility relative to the gas of interest. Most critically, this design allowed multiple iterations to be quickly fabricated and tested. Channels were all 1 mm × 1 mm with an overall path length of 192 mm (see Supplementary Figures 1–3). To measure gas flux rates, an argon sweep gas was used to decrease the temporal delay in gas measurement due to diffusion limits. Absolute concentration measurements were made in the absence of the sweep gas. The release trigger consisted of an electric heating pad set to 65 °C. To deliver localized heating and increase output, we employed a photothermal release mechanism which uses carbon black nanoparticles suspended in the gas capture fluid to drive gas release when exposed to light. Characterization of the photothermal release behavior, characteristics of carbon black and efficiency have been previously reported[5, 18, 19]. Incorporating the directed photothermal release into the counter-current architecture enabled a higher release rate of $CO_2$ from the system. Since each gas requires different carrier solutions, release conditions and measurements, each will be discussed separately. To determine the effects of counter-current amplification and the

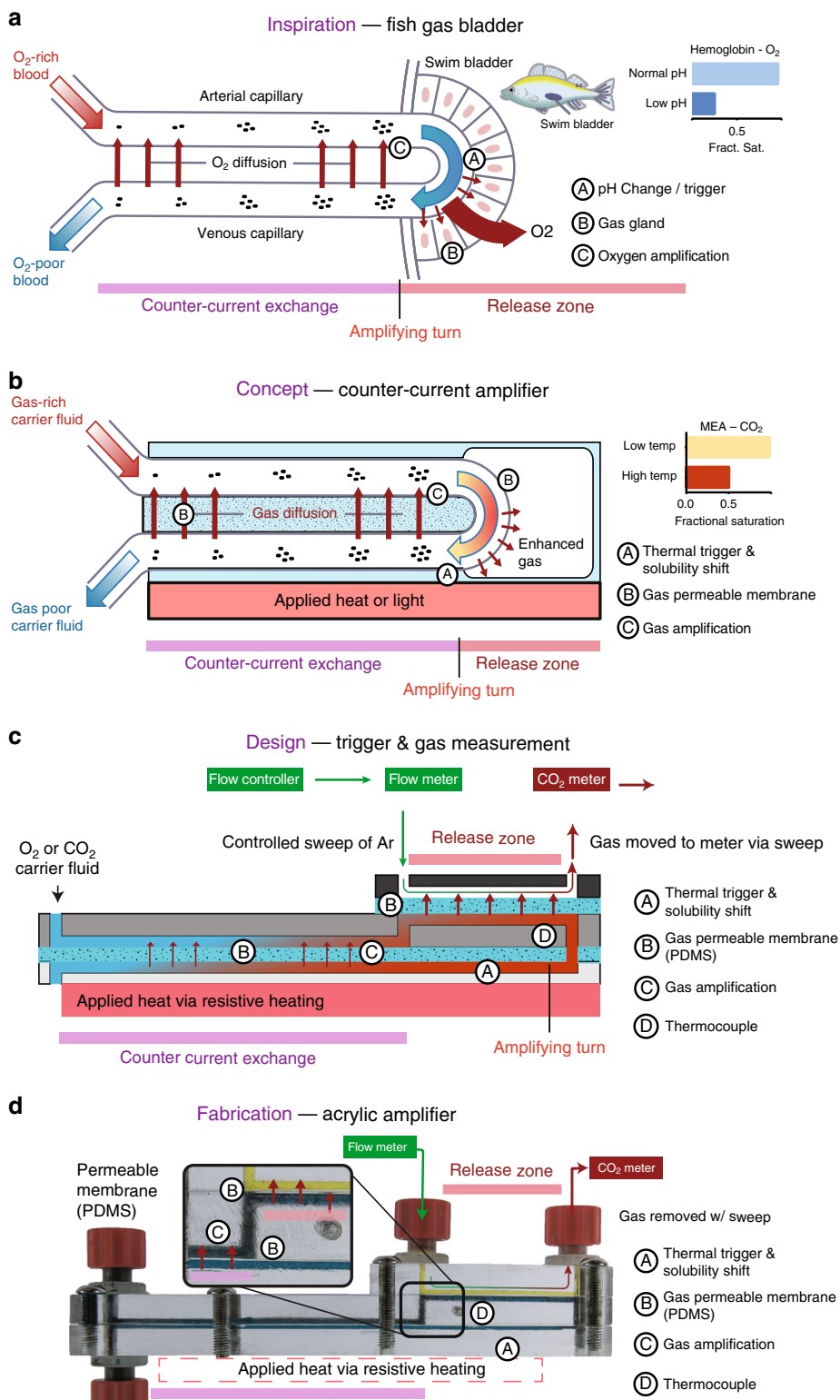

**Fig. 1** Concept and design of the counter-current amplifier system. **a** Diagram of the fish $O_2$ concentration mechanism. A = location of the lactic acid gland used to trigger the release of $O_2$ from hemoglobin as a result of a shift in pH. B = gas gland composed of a semipermeable layer of cells at the interface of the blood vessels and the swim bladder. C = back diffusion of $O_2$ into the inlet channel leading to a concentration amplification. **b** Conceptual diagram of the artificial concentration mechanism. A = location of the thermal or light-based trigger for gas release from the carrier solution. B = location of the gas permeable membranes allowing the diffusion of gas between fluid channels and between fluid and gas channels. C = back diffusion of gas into the inlet channel, leading to concentration amplification. **c** Design of the counter-current amplification device used in this paper. **d** Photo of the fabricated device consisting of 3 layers of acrylic with machined channels and PDMS membranes between the layers

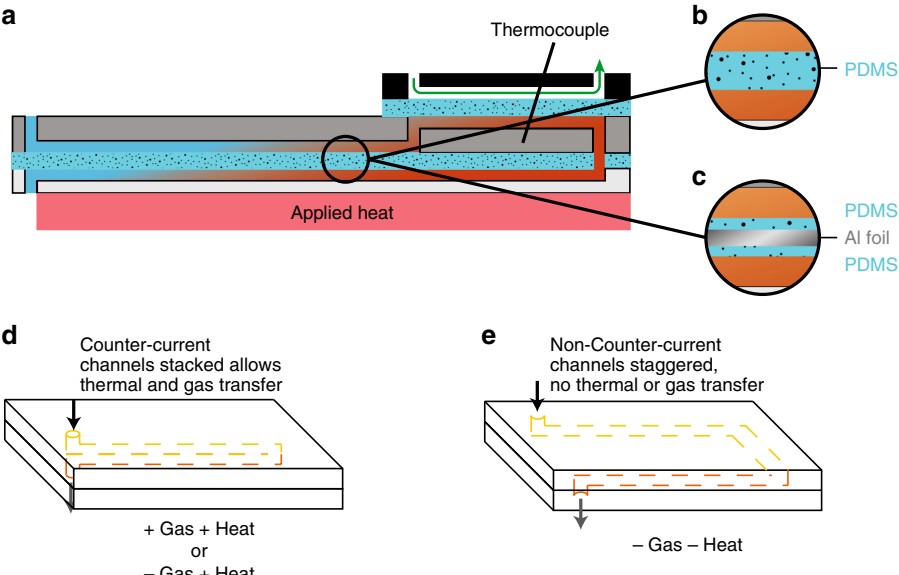

**Fig. 2** Design of experimental controls. **a** Diagram of the counter-current amplification device. **b** Single 140 μm PDMS membrane—allows the transfer of gas and heat between channels. **c** Aluminum foil sandwiched between two 70 μm PDMS membranes—allows the transfer of heat, but not gas. **d** Channel setup for (+Gas +Heat) and (−Gas +Heat)— the incoming (yellow) and outgoing (orange) channels are adjacent, allowing diffusion to take place. **e** Channel setup for (−Gas −Heat)—the incoming (yellow) and outgoing (orange) channels are shifted 5 mm laterally, preventing mass or thermal transfer between them. Note: the dimensions are not to scale and certain components are not shown for clarity

difference between the transfer of heat and the transfer of gas, we validated the counter-current amplification mechanism using monoethanolamine (MEA)/$CO_2$ system as well as exploring $O_2$ separation as a potential application (Fig. 2a–d)[18, 19,]

**Counter-current amplification validation**. For many industrial processes, gas separations make up a large fraction of overall costs[20]. Carbon dioxide is an important precursor chemical to many synthetic feedstocks and plays an increasingly important role in the global climate[21–24]. The industry standard for $CO_2$ capture uses aqueous amines to chemically bind the $CO_2$ combined with a pressure- or thermally driven stripping process to transfer it to storage. The most predominant of these amines is MEA, which has been studied extensively[2, 25–27]. Within the temperature range of this study (25–65 °C, see Supplementary Note 3), the standard MEA:$H_2O$ 30:70 (w/w) solution has a temperature-dependent change in bound $CO_2$ of −0.0048 M K$^{-1}$ [26]. Improving this process of $CO_2$ capture and reducing costs could enable more widespread use of $CO_2$ capture, reducing overall emissions. However, to establish the basic response and critical parameters of an engineered counter-current amplifier, we have explored multiple applications.

For all initial experiments, the carrier fluid flow rate was 1 ml min$^{-1}$. Carbon dioxide was collected with argon as a sweep gas with a flow rate of 5 ml min$^{-1}$, and an applied surface temperature of 65 °C. Use of a sweep gas reduces the time from gas release to gas detection. See Fig. 3a for an example of the experimental output. Under these conditions, $CO_2$ was released at a rate of 6.4 μl min$^{-1}$ (80.2 mmol m$^{-2}$ min$^{-1}$) with an internal temperature of ≈40 °C (Fig. 3b). This rate represents the contributions from counter-current diffusion of gas and heat in the counter-current exchange zone of the swim bladder in addition to the rate expected in the absence of the counter-current exchange zone.

To isolate the contributions of the heat and gas, we fabricated multiple control devices to separate counter-current heat and gas amplification. We altered the counter-current exchange zone by blocking: (i) gas and heat—achieved by diverting the returning channel, eliminating the gas and heat exchange (−Gas −Heat); or

(ii) eliminating the gas exchange between the fluid channels, but allowing heat (−Gas +Heat)—achieved by placing a small layer of aluminum foil within the membrane (Fig. 2a, d).

To test how much of the contribution came from the counter-current amplification rather than simply the applied heat, we built a gas bladder where the incoming and outgoing channels were laterally shifted—removing both heat and gas diffusion between the channels (−Gas −Heat) (Fig. 2e). Under identical heating conditions, the laterally shifted (−Gas −Heat) channels released 2.6 μl min$^{-1}$ (33.2 mmol m$^{-2}$ min$^{-1}$) of $CO_2$ with an internal temperature within the counter-current loop of ≈40 °C. Comparing these data with the full counter-current system (+Gas +Heat) shows a 3.8 μl min$^{-1}$ (59 %) reduction in $CO_2$ release (Fig. 3b). This decrease can be directly contributed to the retention of heat and diffusion of gas due to the counter-current architecture of the counter-current exchange zone.

To isolate the contribution from counter-current gas transfer between the fluid channels from the combined heat and gas transfer, we fabricated a device with a membrane modified to prevent gas diffusion between the fluid channels, while allowing heat (−Gas +Heat). The membrane between the fluid channel and the sweep gas channel was unchanged. The membrane consisted of two 70 μm PDMS membranes on either side of a 20-μm sheet of aluminum foil. Keeping the overall PDMS thickness, the same results in a membrane with a calculated 0.02% increase in thermal resistivity compared to the original 140-μm thick PDMS membrane (Supplementary Note 2). This resulted in essentially no change in the counter-current heat exchange taking place in the counter-current exchange zone, while fully eliminating gas diffusion in that zone (−Gas +Heat). Under otherwise identical conditions, $CO_2$ was released at a rate of 4.3 μl min$^{-1}$ (53.9 mmol m$^{-2}$ min$^{-1}$) with an internal temperature of ≈40 °C (−Gas +Heat). Compared to the full counter-current system (+Gas +Heat), the (−Gas +Heat) system showed a 33% reduction in $CO_2$ release. The difference between the (+Gas +Heat) and the (−Gas +Heat) system can be attributed to the elimination of gas diffusion in the counter-current exchange zone (Fig. 3b).

To achieve precise measurement for our controls, we swept the release zone gas channel with a slow, continuous flow of Argon.

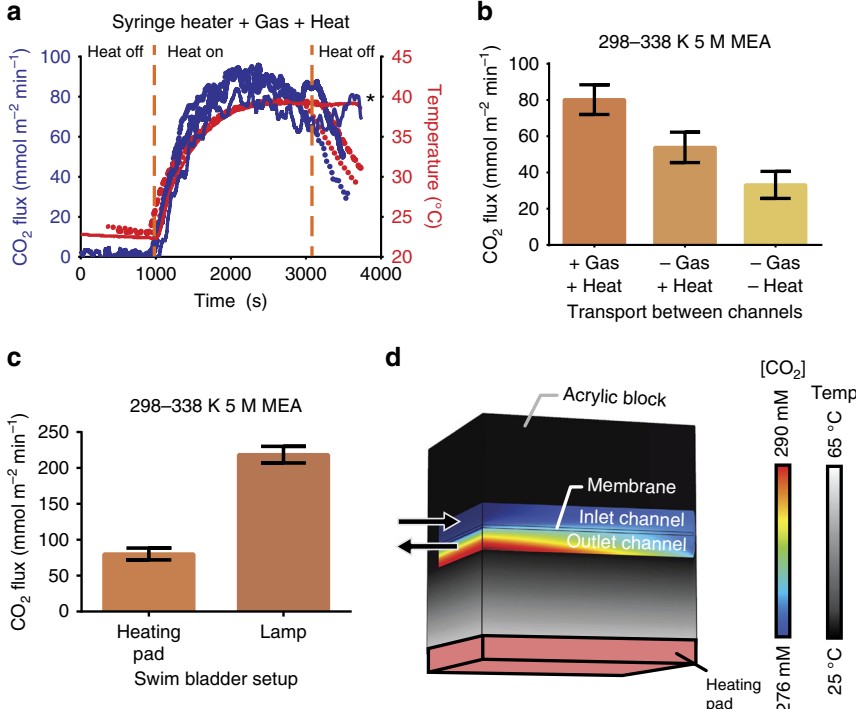

**Fig. 3** Carbon dioxide results and COMSOL model. **a** Representative data output from the counter-current amplifier showing three separate trials (Blue = $CO_2$ flux, Red = temperature). * Run marked with an asterisk was allowed to continue beyond the normal shut off time. **b** Comparison of $CO_2$ release rates for the full counter-current (+Gas +Heat), partial counter-current (−Gas +Heat), and no counter-current (−Gas −Heat) systems. **c** Comparison of $CO_2$ release rates with a heating pad vs. a 300 W lamp. **d** Simplified COMSOL model of the counter-current amplifier, scaled down along the length of the channel for clarity. Gray scale portion indicates the temperature in the acrylic section, and the rainbow scale portion indicates [$CO_2$] in the channel and membrane

An argon sweep is commonly used in the Wicke–Kallenbach method of measuring permeation to enhance diffusion across the membrane and achieve higher rates of gas transfer. The flow rate of gas leaving the system will be higher since a larger gradient between gas and fluid channels is maintained. A sweep gas also prevents the system from reaching its maximum equilibrium concentration, which is of interest for gas concentration applications. A gas-tight adapter for the $CO_2$ meter used in previous experiments was designed and fabricated that allowed the sensor to be directly attached to the counter-current amplifier. Fluid flow rates remained at 1 ml min$^{-1}$, while the sweep gas was eliminated. Gas diffuses into the chamber of the sensor adapter until equilibrium between the fluid phase and gas phase is reached. The head space of the sensor unit is initially filled with ambient air which accounts for the remainder of the gas composition at the experiment end. Without the sweep gas, the gas release rate was not directly measured, and a derived initial flow rate was not considered accurate enough to be reported. The device was preheated to the desired experimental temperature in the absence of the carrier fluid. After verifying that the measured $CO_2$ concentration read by the sensor was below 0.05 atm, the carrier fluid was introduced to the system and flow was initiated. The experiment was run until no significant change in $CO_2$ concentration was observed for 5 min. With a measured internal temperature of ≈35 °C, a maximum concentration of 0.68 atm $CO_2$ was observed. With a measured internal temperature of ≈23 °C (no heating), a maximum concentration of 0.30 atm was observed (see Methods). At an internal temperature of 40 °C, a maximum concentration of at least 1.0 atm $CO_2$ was observed.

**Incorporation of photothermal heating.** While applying thermal heating demonstrated the principles of the counter-current amplification, we sought to increase the output of the device by maximizing the localization of triggering and diffusion. By localizing a heating source to within only the exchange section, we hypothesized we would see a much larger increase in counter-current amplification and gas production. In previous work, we have implemented photothermal heating of nanoparticles to trigger gas release at the nanoscale. This unique effect has been shown by others to produce steam and vapor from water while only heating the solution to 70 °C[5]. This effect works by predominately heating the solution in direct contact with the nanoparticles in solution. By limiting bulk heating, more of the absorbed energy goes into gas release vs. heating the solution. In our own work, we have shown that the photothermal heating of carbon black nanoparticles was sufficient to strip $CO_2$ from MEA/$H_2O$ in bulk or in a 100 μm diameter micro-fluidic channel with a fluid temperature of 65 °C rather than the normal 120 °C. This localized heating and high release rate fit the criteria for our local triggering mechanism[19], so we implemented photothermal release of $CO_2$ mediated by carbon black with only the releasing, outgoing channel replacing the applied heat pad (Fig. 1d). To stimulate the photothermal release of $CO_2$, a 300-W halogen light source was applied at the face of the device with and intensity of 1500 W/m$^2$. In this device, the transparent acrylic top allowed transmission of the light into the nanoparticle-rich solution. The $H_2O$/MEA capture fluid now contained carbon black nanoparticles at 0.2% (w/w) (see Methods) which did not significantly change the amount of $CO_2$ being moved, but provided a photothermal release source. This particle loading also blocks the transmission of light to the incoming channel—meaning minimal light penetrated the back/incoming channel preventing premature release of gas in the counter-current exchange zone (Fig. 1d). With just the applied light, a release rate of 17.35 μl min$^{-1}$ (218.5 mmol m$^{-2}$ min$^{-1}$) was observed with an internal temperature of ≈40 °C, an increase of 170% over the heat-pad triggered system

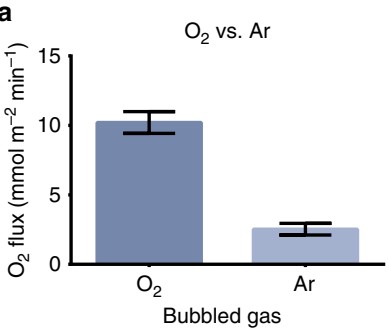

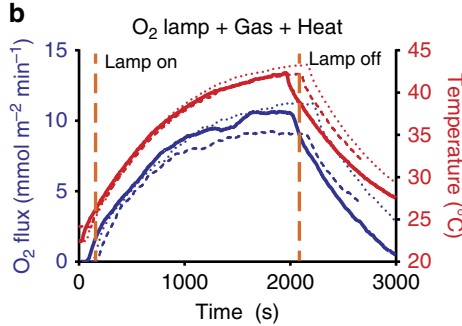

**Fig. 4** Oxygen results. **a** Comparison of oxygen flow with oxygen or argon bubbled into the capture fluid reservoir. **b** Representative oxygen data output from the counter-current amplifier showing three separate trials (Blue = $O_2$ flux, Red = temperature)

(Fig. 3b). Calculating the efficiency of the system based solely on the energy entering channel system, either calculated with COMSOL in the case of the heating pad, or estimated from the surface area of the channels and the intensity of the lamp in the case of the photothermal system gives efficiencies of 1500 MWh tonne$^{-1}$ $CO_2$ and 80 MWh tonne$^{-1}$ $CO_2$, respectively.

**Counter-current amplification of oxygen**. With the counter-current amplification mechanism validated using the $CO_2$/MEA system, we tested the viability of $O_2$ release using perfluorooctane (PFO) as the carrier solution. Rapid access to concentrated $O_2$ in trauma scenarios plays a key role in reducing patient mortality and is necessary for many industrial processes including welding and production of steel[28, 29]. Expanding the breadth of available methods for oxygen concentration beyond existing cryogenic distillation and vacuum swing adsorption could expand the situations and applications where access to oxygen is available. As the gas bladder of fish already purifies oxygen, we engineered our synthetic gas bladder for $O_2$ enhancement. To keep the same photo/thermal triggering mechanism and apparatus, but test $O_2$ enhancement, we employed perfluorooctane as the carrier fluid which has a temperature-dependent change in oxygen solubility of $-0.00013$ M K$^{-1}$ within the temperature range of this study (25–65 °C).

Oxygen amplification presents an additional challenge over $CO_2$, as the change in dissolved gas for a given temperature change of $O_2$/PFO is 2.7% of the $CO_2$ system, while the permeability of $O_2$ through PDMS is 21%, respectively, of the $CO_2$/MEA system[17, 30–32]. The counter-current amplifier configuration was adjusted to accommodate the flow and detection of the lower gas release rate. PFO was saturated with $O_2$ (0.02 M) prior to the start of the run and the flow rate of argon sweep was decreased from 5 ml min$^{-1}$ to 1 ml min$^{-1}$ to accommodate the expected decrease in gas release rate. The $O_2$ percentage was measured using a PreSens Microx 4 fiber optic sensor. All other conditions were kept the same, save for the modification of the carbon black nanoparticles with 1H,1H,2H,2H-perfluorodecyl-trimethoxysilane to improve the suspension within the PFO (see Supplementary Figures 12 and 13)[33]. The release rate of $O_2$ was 0.8 µl min$^{-1}$ (10.2 mmol m$^{-2}$ min$^{-1}$) for the PFO/$O_2$ system—8 times less than the comparable experiment with the MEA/$CO_2$ system (Fig. 4—see Fig. 4b for an example of the experimental output). This result showed that the counter-current amplification system can be extended to other gases where the thermal change in solubility can be induced within a carrier liquid. However, it also shows the importance of overall solubility of the desired purifying gas and the change in solubility upon triggering a change in solubility.

**COMSOL model**. With the encouraging initial results of a counter-current amplifier working for many gases, we sought to

**Table 1 Comparison of the maximum $CO_2$ concentration (initial concentration 0.276 M) in the looped section of the COMSOL model for channel depths of 1.0 mm and 0.1 mm**

| Channel depth (mm) | [$CO_2$] (M) |
| --- | --- |
| 1.0 | 0.282 |
| 0.1 | 0.480 |

examine which parameters would lead to higher yields of gas. We developed a simplified COMSOL model to test how solubility changes, channel geometry, and gas solubility could be used to improve performance of a counter-current amplifier (Fig. 3d). A major difference between the counter-current amplification found in fish and our bio-mimetic system is the diameter of the channels. In the biological system, the channel diameter ranges from 0.001 to 0.01 mm compared to the 1 mm channel depths in our test system. The importance of channel size has been noted in various theoretical treatments of the swim-bladder system[13, 14, 34]. Using a simplified COMSOL model, we saw that decreasing the channel depth to 0.1 mm from 1 mm resulted in a 34-fold increase in the difference between the initial concentration and the concentration under active heating (see Table 1 and Supplementary Figure 20). The shorter average diffusive path between inlet and outlet channels allows for a greater proportion of released gas to be recycled into the inlet channel before exiting the system[14]. Our current design allows the device to be easily scaled using common fabrication techniques including laser etching, or CNC machining in addition to pursuing the use of hollow-fiber membrane contactors[35]. In future work, we plan to tackle the engineering challenges required to create counter-current amplifiers more closely matching those seen in nature.

**Discussion**
Here, we fabricated a counter-current amplifier using micro-machined acrylic components for the inlet and outlet channels and PDMS membranes for a semipermeable membrane (Fig. 1d). The change in solubility was triggered using a thermal and photothermal effect that enabled changes in solubility along the axis of counter-current exchange. We generalized the counter-current system to release both $CO_2$ and $O_2$ by switching the carrier fluid and matching each carrier fluid with a thermal switch of solubility. Using control devices, we observed a 240% improvement with our counter-current amplification system (+Gas +Heat) vs. the sweep gas only system (−Heat −Gas) and were able to isolate the contributions of thermal and gas diffusion to the enhancement. Considering the counter-current transfer of heat between channels, the total improvement due to the counter-

current diffusion of $CO_2$ was 150%. Notably, in our best performing systems, the release mechanism was provided by simulated sunlight at ≈3 times typical solar irradiance. Photothermal driven release of gases opens the door to direct solar driven concentration. Shifting toward the use of higher efficiency LED lights in addition to work by Halas et al. demonstrating increased efficiency in cross-membrane transport using nanoparticles isolated to the surface of the membrane present clear avenues toward optimizing the system[36].

Our results suggest that counter-current amplification is a general principle that can be applied to many different heat-driven gas separation challenges. For $CO_2$, a counter-current amplifier could be used to lower the energy cost of the stripping process of $CO_2$ associated with carbon capture operations. For $O_2$, both the improved release rate and smaller footprint might improve oxygen delivery for combat and point-of-care treatment. In these applications, the lower power and smaller volume will be critical to making a more portable oxygen amplifier which could increase $O_2$ levels to near medical grade from atmospheric oxygen.

These exciting applications will require overcoming challenges and making improvements to counter-current amplifiers. In this work, we constructed a model to examine how improvements in counter-current amplification could be implemented. Notably, the rate of release and rate of permeation of the membrane should yield increases in counter-current amplification. In particular, the membrane permeability can be rapidly adjusted by using semi-porous membranes that should allow direct $CO_2$ diffusion through liquid channels. The largest gain in performance can be achieved in size of individual channels, which is currently 1.0 mm, if lowered to 0.1 mm, could yield improvements of 35-fold in concentration increase due to the release trigger. Our intention is to develop counter-current amplifiers with smaller channels that are interdigitated—analogous to our work on biomimetic exchangers[37, 38]. These smaller channels should yield the desired improvements in total gas released and gas purity as hinted at by the COMSOL work.

In its current form, our device is several orders of magnitude less efficient than existing commercial carbon sequestration methods (80 MWh tonne$^{-1}$ $CO_2$ vs. 0.35 MWh tonne$^{-1}$ $CO_2$). Use of heat exchangers coupling the absorption step with the release stage allows a significant reduction in the energy required for the entire cycle. Future work will focus on integrating heat exchange between absorption and desorption and minimizing the heat loss from the face opposite the heating device. The device, as it is, serves as a proof-of-concept application of the counter-current amplification motif seen in fish by demonstrating how a simple architectural change can lead to an increase in the output of a gas desorption device. In addition, larger changes in solubility are needed to achieve higher efficiencies and lower footprints. Photothermal heating mediates phase separation as well as gas release and we believe that a combination of solvents with different solubilities can be used to enhance the change in solubility for our photothermal triggering system[39].

Beyond these challenges and improvements, the potential of these systems to create large concentration gradients may open the door to a variety of fields where a solute is concentrated from a dilute initial solution. We plan to apply this work to additional solutes and release triggers and are currently investigating the use of this technique for heavy metal cleanup, concentration of proteins, and isolation of other dilute gases.

## Methods

**Fabrication of counter-current amplifier**. All counter-current devices were machined from 5-mm thick acrylic sheets purchased from McMaster-Carr. Fluid channels are all 1-mm wide by 1-mm deep square cross-sections for ease of fabrication. Technical drawings of the channel layouts can be found in Supplementary Figures 1 and 2.

**Fabrication of hybrid membrane**. A hybrid membrane was designed and fabricated as illustrated in Supplementary Figure 5 consisting of three layers. The two outer layers of PDMS maintain a consistent thermal resistivity compared to the original PDMS membrane as well as preventing corrosion of the aluminum layer by the MEA solution. The aluminum foil serves as a gas impermeable layer. Calculations detailing the thermal conductivity of the hybrid membrane compared to the PDMS membrane can be found in Supplementary Note 2.

**Experimental setup**. The gas detection system was designed to pick up gas flow rates in the 0.1–50 μl range. A photo of the experimental setup can be seen in Supplementary Figure 6. A Smart-Trak 50 mass flow controller was used to ensure a consistent flow of argon. A Cole-Parmer mass flow logger (60 data points/second) and CO2Meters.com $CO_2$ sensor or PreSens $O_2$ sensor (1/3 Hz sampling frequency) enabled the release rate of the gas of interest to be measured. The flow meter was located upstream of the point of gas release due to the sensor's sensitivity to changes in gas composition. The additional volume of gas released by the counter-current amplification system was considered negligible compared to the flow rate of the sweep gas. Flow rate data and percent gas data were combined in Mathematica by interpolating the data with a moving average of 20 points for the concentration data and 1000 points for flow rate data.

**Oxidation of carbon black**. Carbon black was oxidized using a modified procedure from Rouxhet and colleagues[40]. Two grams of carbon black (Cabot, N115) and 60 ml of 30% $H_2O_2$ (VWR, 30%) were added to a 125-ml round bottom. The round bottom was fitted with a condenser and stir bar and brought to reflux for 24 h. The carbon black was separated from the $H_2O_2$ peroxide by filtration, washed with water, and dried in a 120 °C oven. Oxidized carbon black was dried in a 110 °C vacuum oven for 1 h.

**Perfluorosilane modification of carbon black**. Carbon black (500 mg) was weighed out into a 20-ml scintillation flask with stir bar charged with 5 ml of 1H,2H,2H,-pefluorodecyltrimethoxysilane. The flask was purged with argon and sealed. The solution was stirred for 24 h after which the solution was filtered and the carbon black nanoparticles were rinsed twice with hexanes and water, then dried in the vacuum oven overnight. Characterization data can be seen in Supplementary Figure 11.

**Carbon black in PFO suspension tests**. The dispersion of the carbon black nanoparticles was measured relative to raw carbon black using optical microscopy. Carbon black was added to 1 ml of PFO at a concentration of 0.13% w/w and then sonicated for 180 s after all additives were added. A drop of the resulting solution was placed on a microscope slide and viewed using a 5× objective.

Images where processed using ImageJ following a modified version of a tutorial from microscopy.ethz.ch[4]. The images were converted to binary, black, and white images by thresholding. The resulting images were processed with the "Analyze Particles" tool resulting the mean, max, min, and standard deviation of the particle area.

Photos of the suspensions can be seen in Supplementary Figure 12, 13 and the ImageJ analyses can be seen in Supplementary Tables 1, 2.

**Carbon black in MEA light transmittance**. Percent transmittances were measured using a Zeiss AXIO Observer.A1 microscope. Carbon black suspensions were placed in a 2-mm-deep acrylic channel sealed with layer of PDMS. The microscope camera was set to an exposure time of 400 μs and the lamp was set to the highest intensity possible without overexposing the RGB color channels. Data for light transmittance can be seen in Supplementary Figure 14.

**Estimation of energy use**. The energy use of the system was estimated by building a COMSOL model of the experimental system accurate in the dimensions given in Supplementary Figure 1 to investigate heat transfer within the model. The power flux (W/m$^2$) of a particular surface can be exported. This is illustrated in Supplementary Figure 21 with a color map where any pixel with a value of 0 (black) has a power flux of 0 W/m$^2$ or less. The integration of that plot with respect to area gives the total power coming into the volume of interest from that face. Combining the integrated values from each face of the channels (Supplementary Figure 22) gives the total power entering the system which can be used to calculate the efficiency of the system—1500 MWh/tonne$_{CO2}$

**Data availability**. The data and simulation codes that support the findings of this study are available from the corresponding author on request.

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

## Acknowledgements
We thank the Office of Naval Research, the Air Force Office of Scientific Research, and the National Science Foundation Center for Chemistry at the Space-Time Limit. This work was supported by ONR (N000141410503), AOFSR (FA9550-15-1-0300, FA9550-16-1-0017), NSFCCSTL (CHE-082913).

## Author contributions
K.B. and A.P.E.-K. devised the research. K.B. designed and implemented the experimental system and developed the data analysis. A.G. provided insight in optimizing the physical setup and data collection process. K.B. and A.G. performed the data collection. R.C.S. gathered EDS data on the carbon black modifications. The manuscript was written by K.B. and A.P.E.-K.

## Additional information

**Competing interests:** The authors declare no competing financial interests.

