## [Peer Review File · Nature Communications]

Reviewers' comments:

Reviewer #1 (Remarks to the Author):

The present manuscript deals with the development of countercurrent amplification method, as a new system for gas purification and enhancement. The novelty of the claims is definitely undoubted and, for sure, this manuscript will have a wide echo in the field. The manuscript is very clearly written and well supported by data; Moreover, it supplies all the information required to reproduce the results. Therefore, I strongly suggest to publish this manuscript in NatComm as it will provide interesting insights to the scientific community.

Two small comments:

- I did not catch very well why a PDMS membrane has been chosen. Can you clarify, please?
- A more general question is on the perspective of using this system in real applications. What will be the behavior (efficiency) of the system with real flue gas streams which contain also impurities like SO_x and NO_x? What can be the limitations and the requirements for really comparing this system with well consolidated gas separation technologies?

Reviewer #2 (Remarks to the Author):

The current manuscript describes a device that can able "countercurrent amplification" of gases for gas purification applications.

The work is very creative and the bio-inspired approach to separations is novel. However, the article and the research is difficult to understand--even to experts in the area--within the context of membrane science or separations.

The article as it stands is not of sufficient quality for Nature Communications nor does it convince the reader that the device is truly a breakthrough relative to existing "amplification" techniques. The article should be clarified, revised, and submitted elsewhere.

Specific technical comments:

1. The authors need to explain how this device is different than typical countercurrent membrane-mediated stripping operations. This is an active area of research, and the authors should use this as a backdrop for their own research.
2. Using an argon sweep on the "permeate" side of the PDMS membrane is reminiscent of the Wicke-Kallenbach membrane permeation technique. This technique provides essentially the maximum driving force for separation under a given concentration of CO₂ on the upstream. Importantly, it is not considered practical at all, as you have taken your concentrated CO₂ (in MEA) and diluted it into argon (now another separation is needed). The Wicke-Kallenbach approach is fine for exploratory research, but the authors need to speculate on how an actual amplification device would actually operate (i.e., how does one recover pure CO₂ from this device?).
3. I believe that the authors are considering this device for the removal of CO₂ from CO₂-loaded MEA streams, which implies that this device would replace "stripping" columns in normal absorber-stripper unit operations. This should be clearer in the article and in the figures.
4. Reporting the data in flow rates (and as a function of feed flow rate) is mostly meaningless, as the reader has no context on the mass transfer area, the driving force for separation, etc. The authors need to report the data in normalized permeability or permeance terms so that their device can be compared to state-of-the-art separation membranes.

5. With comment #4 in mind, the authors need to make explicitly clear what advantage their device brings to gas purification applications that cannot be addressed by other materials or technologies.

6. I do not think the loading of CO₂ and O₂ in solvents is linear with temperature. Gibbs-Helmholtz and van't Hoff relations suggest that it is not linear. In fact, the rate equations in the SI also show this, I believe.

7. Figure 1 is not clear. In (a), what is the stream flowing into the swim bladder? Where is the O₂ going? Similar questions for (b). In (e), it appears as if the flow controller and flow meter are "flowing" into the amplification device (i.e., what are the green arrows?).

8. Although this is just a proof-of-concept, most membrane devices try to drive down device volume by maximizing mass transport area within the device. State-of-the-art fiber systems can approach 10,000 m²/m³. Based on the introduction, it appears that the authors insinuate that their devices could eventually reach mass transport areas in excess of this, but based on the designs shown in Fig. 1f it is not clear how that would be achieved. The authors should clarify this point.

9) Inspecting Figure 1e and 2a, it is not clear where the released CO₂ is going? Especially in figure 2b, where an aluminum foil sheet is sandwiched between two PDMS layers---how does the CO₂ leave the device in this case? The authors need to clarify this point.

10) Figure 3c is not clear at all.

11) There are many typos throughout the manuscript, and the writing could be improved substantially.

Overall, this is a very creative effort, but significant work is needed before it is ready for publication. I do not recommend it for publication in Nature Communications.

Reviewer #3 (Remarks to the Author):

In their work, Esser-Khan and colleagues realize an artificial countercurrent amplification system with potential applications in gas enrichment and/or solutes concentration processes.

Countercurrent mechanisms are very important procedures happening in nature which employ energy to induce the formation of concentration gradients facilitating, for example, the extraction of concentrated solutes from a watery environment. From a more general point of view the principle can be applied as a separation method and find applications in different contexts such as gas concentration, as suggested by the authors.

While the idea of realizing an artificial device for multipurpose gas separation is relevant, the manuscript at this stage cannot be published as it lacks of a structure and unfortunately is poorly written.

In the following some of the comments which may help the authors for an improved version of their work:

1. General structure: the work is about countercurrent amplification but a clear and basic description of its working mechanism is not clearly provided. Some examples are given throughout the text but an explicit explanation is missing. I think the reader would benefit from it. In particular:

a. The focus: is gas separation or solute concentration in general the target of the work? Artificial processes for solutes concentration have been realized (e.g. research on artificial kidneys) so I would narrow the potential applications of the proposed system if the claim to be the first of its kind must hold.

b. What are current technologies and how the new system would improve the situation: it is claimed that 10-15% of world's energy consumption is related to gas concentration. What are current requirements for gas separation to be employed on the market in terms of efficacy of separation, type of gases that can be separated, costs etc.. ? And more importantly, how the new proposed technology would improve the situation? How the 10-15% energy consumption would be lowered? Carbon black is mentioned as a method to deliver localized heating. In fact it has been used to efficiently convert sunlight into vapor but in the manuscript renewable solar irradiation is not directly mentioned. A lack of carbon black absorption analysis is also missing.

c. Main claim: it is mentioned that a 246% increase in the rate of gas release has been achieved by means of countercurrent amplification. However this number is not reported directly in the abstract and it is not clear what its actual relevance is. Should this value be compared to something? It is not clear what it stands for. Is it the main quantitative claim of the work? If yes I would mention it way earlier in the text but immediately providing a comparison which shows what the increase is referred to.

d. Results and discussion: the results section lacks of a qualitative description of the experiments. Key points are not highlighted. A lot of details and numbers are given but no accent is put on the major drivers of the technology. Discussion part sound only like a conclusion where findings are summarized.

2. Figures: most of the figures are not adequately explained and some panels are not even mentioned in the text.

a. Fig 1: only fig 1 a and c are mentioned in the text. b,d,e and f panel are missing in the main text so that their relevance is not clear. Panel c is not described properly and lacks of details.

b. Fig2: only panel d is mentioned in the text. All the other panels are missing.

c. Fig 3c: not mentioned.

3. Supplementary Materials: it is not clear how Supplementary Materials should be used. Besides the different formats in which they appear, they lack of basic explanations and they are often not even mentioned in the text. I would suggest giving them a structure to highlight how they can be used to support what is presented in the main text.

4. Readability: the writing style is often confusing. Plus there are several typos and errors throughout the text.

Since the overall idea can be of interest to the scientific community, I encourage the authors to restructure the paper and present the results in an improved manuscript.

Reviewer 1

Comments:

1. I did not catch very well why a PDMS membrane has been chosen. Can you clarify, please?
 - a. We thank the reviewer for pointing out that this information was missing. We have added details on the choice of membrane material in paragraph 5.
 - i. “Polydimethyl siloxane (PDMS, 140 μm thick) was chosen as the membrane material due to the ease of fabrication of varying thicknesses and the relatively high permeability of both CO_2 and O_2 (3800 and 800 Barrer respectively) through it.¹⁹”

2. A more general question is on the perspective of using this system in real applications. What will be the behavior (efficiency) of the system with real flue gas streams which contain also impurities like SO_x and NO_x ? What can be the limitations and the requirements for really comparing this system with well consolidated gas separation technologies?
 - a. We appreciate the forward-looking comments here and wish we could address them all in one MS. Our goal in this paper was primarily to demonstrate an increase in gas output for a given energy input, but we recognize the challenges the system faces in real world applications. We have amended the text in paragraph 5 to clarify that the effect of countercurrent amplification on contaminants is unclear and dependent on the relative temperature dependent solubilities.
 - i. “Gas selectivity in our test system was driven by the carrier fluid and the source gas, and the effect of this system on contaminant gas concentration was not investigated. The behavior of contaminants in our system likely depends on the change in solubility relative to the gas of interest.”

Reviewer 2

Comments:

1. The authors need to explain how this device is different than typical countercurrent membrane-mediated stripping operations. This is an active area of research, and the authors should use this as a backdrop for their own research.
 - a. We agree. This point was not well clarified in the original text and apologize for the confusion. We have included discussion in paragraph 1 on how traditional countercurrent membrane strippers are limited to an output concentration equivalent to that of the partial pressure above a bulk solution at the release temperature, while the countercurrent amplification system can achieve higher concentrations for the same conditions.
 - i. "For traditional countercurrent extraction, the output concentration is limited to the partial pressure of the gas above a bulk phase of the carrier fluid at the release conditions used. In contrast, countercurrent amplification allows the output concentration to reach many times that of bulk phase partial pressures."
2. Using an argon sweep on the "permeate" side of the PDMS membrane is reminiscent of the Wicke-Kallenbach membrane permeation technique. This technique provides essentially the maximum driving force for separation under a given concentration of CO₂ on the upstream. Importantly, it is not considered practical at all, as you have taken your concentrated CO₂ (in MEA) and diluted it into argon (now another separation is needed). The Wicke-Kallenbach approach is fine for exploratory

research, but the authors need to speculate on how an actual amplification device would actually operate (i.e., how does one recover pure CO₂ from this device?).

- a. We appreciate the concern here and think it's a valid criticism. We did further experiments to directly address this question. The results, we believe, have greatly strengthened the MS. We originally implemented the sweep gas because we wanted a very simple measurement of the gas coming out and to minimize error, and have clarified this in paragraph 5.
 - i. "Channels were all 1 mm x 1 mm with an overall path length of 192 mm. To measure flux rates, an argon sweep gas was used to decrease the temporal delay in gas release and gas measurement due to diffusion limits."
 - b. Recognizing that a sweep gas is not practical for real world applications, we also measured the equilibrium gas concentration without a sweep gas. This information is detailed in paragraphs 5 and 12. We were very gratified to see that the gas bladder performed as expected and we saw that at 40 C we could measure 100% CO₂ being admitted from the device.
 - i. "Absolute concentration measurements were made in the absence of the sweep gas."
 - ii. "With a measured internal temperature of ≈ 35 °C, a maximum concentration of 68 % CO₂ was observed. With a measured internal temperature of ≈ 23 °C (no heating), a maximum concentration of 30 % was observed (SI sec. 2.1). At an internal temperature of 40 °C, a maximum concentration of at least 100 % CO₂ was observed."
3. I believe that the authors are considering this device for the removal of CO₂ from CO₂-loaded MEA streams, which implies that this device would replace "stripping" columns in normal absorber-stripper unit operations. This should be clearer in the article and in the figures.
- a. We appreciate the comments and have added clearer language describing potential future applications of the countercurrent amplification mechanism in paragraph in paragraph 17. Our long-term vision is that this might be used in stripping experiments, but for this publication we wanted to focus on the fundamental design and testing – to publish the principle rather than focus on an application. We amended the text to read:
 - i. "Our results suggest that countercurrent amplification is a general principle that can be applied to many different heat driven gas separation challenges. In CO₂, a countercurrent amplifier could be used to lower the energy cost associated with stripping CO₂ during the

capture and release process. The $\approx 242\%$ improvement in CO₂ released for an equivalent amount of heat might lower energy costs. One area that might benefit is the stripping of CO₂. “

4. Reporting the data in flow rates (and as a function of feed flow rate) is mostly meaningless, as the reader has no context on the mass transfer area, the driving force for separation, etc. The authors need to report the data in normalized permeability or permeance terms so that their device can be compared to state-of-the-art separation membranes.
 - a. We thank the reviewer for this comment. The data has been reprocessed in units of $\text{mmol}\cdot\text{m}^{-2}\cdot\text{min}^{-1}$, which is more consistent with existing mass transfer studies.
5. With comment #4 in mind, the authors need to make explicitly clear what advantage their device brings to gas purification applications that cannot be addressed by other materials or technologies.
 - a. We apologize for not making this clear. We believe this point is addressed in our response to comment 1, where we discuss the differences in equilibrium concentration between existing countercurrent strippers and the countercurrent amplification system.
6. I do not think the loading of CO₂ and O₂ in solvents is linear with temperature. Gibbs-Helmholtz and van't Hoff relations suggest that it is not linear. In fact, the rate equations in the SI also show this, I believe.
 - a. We thank the reviewer for these thoughts. The solubility of CO₂ and O₂ in their respective carrier fluids is not linear over all temperatures, but can be approximated as such over the temperature range in our study. We believe a simpler linear relationship is easier for the reader to translate into concentration changes than a more complex relationship. We have added text clarifying this point in paragraphs 3, and 4. Details regarding the derivation of the solubility relationship can be found in the section 4.2 of the supporting information.
7. Figure 1 is not clear. In (a), what is the stream flowing into the swim bladder? Where is the O₂ going? Similar questions for (b). In (e), it appears as if the flow controller and flow meter are "flowing" into the amplification device (i.e., what are the green arrows?).
 - a. We thank the reviewer for their input. We have adjusted the figures to make them clearer about what is going on. In (a), the stream flowing into the

channels is blood, and oxygen diffuses into the swim bladder gas chamber. We have modified the image to clarify this.

- b. 1b has been modified in the same manner as 1a
 - c. What was figure 1e has been changed to figure 1c. The flow controller pictured is for the sweep gas used in the experiments. We have modified the figure to clarify this and have added text explaining the relative positions of the components in the “Experimental Setup” section under “Methods”.
 - i. “The flow meter was located upstream of the point of gas release due to the sensor's sensitivity to changes in gas composition. The additional volume of gas released by the countercurrent amplification system was considered negligible compared to the flow rate of the sweep gas.”
8. Although this is just a proof-of-concept, most membrane devices try to drive down device volume by maximizing mass transport area within the device. State-of-the-art fiber systems can approach 10,000 m²/m³. Based on the introduction, it appears that the authors insinuate that their devices could eventually reach mass transport areas in excess of this, but based on the designs shown in Fig. 1f it is not clear how that would be achieved. The authors should clarify this point.
- a. We have included text describing our intent to miniaturize and parallelize the existing design in paragraph 14.
 - i. “Using a simplified COMSOL model, we saw that decreasing the channel depth to 0.1 mm from 1 mm resulted in a 51 fold increase in the difference between the room temperature concentration and the concentration under active heating. The shorter average diffusive path between inlet and outlet channels allows for a greater proportion of released gas to be recycled into the inlet channel before exiting the system.¹⁶ Our current design allows the device to be easily scaled using common fabrication techniques including laser etching, or CNC machining.³⁷ In future work, we plan to tackle the engineering challenges required to create countercurrent amplifiers more closely matching those seen in nature.”
9. Inspecting Figure 1e and 2a, it is not clear where the released CO₂ is going? Especially in figure 2b, where an aluminum foil sheet is sandwiched between two PDMS layers---how does the CO₂ leave the device in this case? The authors need to clarify this point.

- a. We have clarified this point in paragraph 11 in addition to modifying the figures to make them clearer. The hybrid membrane is only located between the fluid-fluid channels, and the membrane between the gas-fluid channel is unchanged.
 - i. “To isolate the contribution from countercurrent gas transfer between the fluid channels from the combined heat and gas transfer, we isolated the countercurrent heat transfer via a second control device. We fabricated a device with a membrane modified to prevent gas diffusion between the fluid channels, but allow heat transfer between incoming and outgoing channels (- Gas + Heat). The membrane between the fluid channel and the sweep gas channel was unchanged.”

10. Figure 3c is not clear at all.

- a. We apologize for the lack of clarity. We have modified the figure to make the construction of the model clearer. Labels have also been added to give the reader signs to understand the layout of the image. The description of the image has been clarified as well.

11. There are many typos throughout the manuscript, and the writing could be improved substantially.

- a. We apologize for these typos and for the general lack of clarity in the first draft. A significant portion of the manuscript has been reorganized and rewritten, and typos have been fixed to the best of our ability.

Reviewer 3

Comments:

1. General structure: the work is about countercurrent amplification but a clear and basic description of its working mechanism is not clearly provided. Some examples are given throughout the text but an explicit explanation is missing. I think the reader would benefit from it. In particular:
 - a. The focus: is gas separation or solute concentration in general the target of the work? Artificial processes for solutes concentration have been realized (e.g. research on artificial kidneys) so I would narrow the potential applications of the proposed system if the claim to be the first of its kind must hold.
 - a. We thank the reviewer for alerting us to this point of confusion. We have clarified the underlying mechanism for amplification in paragraph 2. We have also updated figure 1 to make the visual description of the effect clearer.
 - i. “The key to attaining these concentrations is a series of looped blood vessels between which gas can diffuse combined with a gas release

trigger. Blood flowing into the swim bladder passes by a loop adjacent to a gas storage bladder where the release of oxygen from hemoglobin is triggered. Now rich in free oxygen, the blood flows back alongside the incoming channel where free oxygen can diffuse into the incoming channel, increasing the concentration of hemoglobin bound oxygen. The cycle repeats, successively increasing the oxygen concentration in the loop until an equilibrium is reached. (Fig 1a)”

- b. We have clarified the application and novelty of the project in the abstract.
 - i. “To assess the potential for counter-current amplification to address modern gas separations challenges, we developed the first synthetic countercurrent amplification system for gas purification and enhancement.”
2. What are current technologies and how the new system would improve the situation: it is claimed that 10-15% of world’s energy consumption is related to gas concentration. What are current requirements for gas separation to be employed on the market in terms of efficacy of separation, type of gases that can be separated, costs etc.. ? And more importantly, how the new proposed technology would improve the situation? How the 10-15% energy consumption would be lowered? Carbon black is mentioned as a method to deliver localized heating. In fact it has been used to efficiently convert sunlight into vapor but in the manuscript renewable solar irradiation is not directly mentioned. A lack of carbon black absorption analysis is also missing.
 - a. The economic crossover point varies widely depending on local subsidies and methods of energy generation. For this paper we are more interested in demonstrating that an appreciable improvement can be made over a system equivalent to the current methods of gas capture. The demonstrated increase in gas release for the same amount of energy should be scalable and result in an overall decrease in energy use for a given capture and release system.
 - b. We have clarified the preexisting work on the photothermal heating effect in our lab and others and included language describing the use of solar simulators for gas release in paragraph 2. Information regarding the absorbance of carbon black in solution has been added to the supporting information in section 3.3.
3. Main claim: it is mentioned that a 246% increase in the rate of gas release has been achieved by means of countercurrent amplification. However this number is not reported directly in the abstract and it is not clear what its actual relevance is. Should this value be compared to something? It is not clear what it stands for. Is it the main

quantitative claim of the work? If yes I would mention it way earlier in the text but immediately providing a comparison which shows what the increase is referred to.

a. We apologize for the lack of clarity. We have modified the text to clarify the origin of the reported percent increase in the abstract and in paragraph 12. It should be considered a 246% increase over simply heating the same flowing solution.

4. Results and discussion: the results section lacks of a qualitative description of the experiments. Key points are not highlighted. A lot of details and numbers are given but no accent is put on the major drivers of the technology. Discussion part sound only like a conclusion where findings are summarized.

a. We apologize. After working on the MS for some time, we struggled to determine how to report our ideas in the most effective manner. We have included more details on the experimental setup and reasoning for the specific choices made for data acquisition.

5. Figures: most of the figures are not adequately explained and some panels are not even mentioned in the text. a. Fig 1: only fig 1 a and c are mentioned in the text. b,d,e and f panel are missing in the main text so that their relevance is not clear. Panel c is not described properly and lacks of details. b. Fig2: only panel d is mentioned in the text. All the other panels are missing. c. Fig 3c: not mentioned.

a. Additional figure callouts have been added to the text to ensure the reader is directed to the correct result in the text.

6. Supplementary Materials: it is not clear how Supplementary Materials should be used. Besides the different formats in which they appear, they lack of basic explanations and they are often not even mentioned in the text. I would suggest giving them a structure to highlight how they can be used to support what is presented in the main text.

a. The SI has been reformatted to match the main text. Additional clarification of the supporting information material has been included.

7. Readability: the writing style is often confusing. Plus there are several typos and errors throughout the text. Since the overall idea can be of interest to the scientific community, I encourage the authors to restructure the paper and present the results in an improved manuscript.

a. A significant portion of the manuscript has been reorganized and rewritten, and typos have been fixed to the best of our ability.

The reviewers offered very helpful comments and suggestions. We have made all necessary changes to clarify any ambiguities within the text and the figures. We hope this satisfies the questions raised in initial reviews.

With these corrections and adjustments, we believe we have satisfied all the reviewers' concerns. We thank them for their careful attention to detail and help in improving our experiments.

Reviewers' comments:

Reviewer #2 (Remarks to the Author):

I thank the authors for their thoughtful responses to my original critiques. These helped improve the article, although there are still some major issues. None of these show-stoppers, and the article should still be considered for eventual publication in Nature Communication.

The short version of my critique is that it appears that the authors do not want to tangibly compare their method to any other method of separating CO₂ from MEA and instead rely on circumstantial discussion to prove their method should be capable of producing purer gas streams than other methods. Indeed, the article would benefit greatly if their new method were to be compared on an energy or area basis. The writing is still difficult to understand throughout the article.

I will go through the responses in detail one by one.

Comment 1. Concerning not discussing countercurrent membrane mediated stripping.

The authors address this well enough in that they at least mention it and explain why their device can offer advantages over other methods circumstantially.

A concern throughout is the fact the authors only seem to benchmark against their own device and not against any kind of standard. I am guessing no such standard exists, but coming up with some comparison metric would substantially help the article.

Comment 2. Concerning Wicke-Kallenbach

The authors measure the concentration of the gas when no sweep is used, but do not report flow rates, only concentration. Somehow a 68% pure stream leaving the system is observed when operating with no sweep at 35 °C. What the other 32% of the stream was is never addressed.

Comment 3. Authors should have some discussion of where their separations device would operate and what it may replace (absorber-stripper operations)

The authors have addressed my concern by added this discussion at the back end of the paper, where it really should have gone up front to help the reader understand the purpose of the work.

Comment 4: OK

Comment 5: See response to Comment 1

Comment 6: OK

Comment 7: OK

Comment 8. How can this device reach similar mass transfer areas to other devices such as hollow fibers?

As discussed in Comment 1, benchmarking this device against anything else would help the paper substantially. The authors argue their Comsol modelling shows that if they were to shrink down the size of the channels they can increase purity in the channel. That said, nowhere in the SI (Section 4.3) do they actually show comparisons between the 1mm channel depth and the 0.1 mm channel depth. The only place we see this reported is figure 3d. The authors really should include the plots comparing concentration of the two channel depths in the SI in more detail.

Comment 9: OK

Comment 10: OK

Comment 11: The writing has been improved, but is still confusing to the reader. Perhaps having an unfamiliar but technically savvy colleague go through the article will help clarify this article.

New comments:

Figures 2e and 4b are never directly referenced or explained in the manuscript, as far as I can tell. Why is the flux dropping off strongly near the end of the experiment? Did the heat/light turn off, and if so, why isn't that noted in the plot.

Incorrect units:

Line 288 The gas detection system was designed to pick up gas flow rates in the 0.1 – 50 μm range. Gas flow rates, length units?

Line 208, units of "rate of O₂ production was 0.8 ml/min" I am almost positive based on context this should be $\mu\text{L}/\text{min}$

SI:

The calculation of thermal resistance (line 57 of the SI) seems to have a typo. Their solution is right, but the value 0.000020 should be 0.000040.

Figure S9. The figures are not labelled A and B, although the captions calls on them as though they are. Also the y-axis on figure S9 Right (B?) likely should read Ar Flux (units) since there is no O₂ in that experiment?

Reviewer #3 (Remarks to the Author):

The manuscript has improved and it is now well written. However there are some points that should be addressed. The main concern is about the energy comparison between the different heating methods.

Main comments are:

1. The 242% increase is mentioned to be the improvement between the light induced experiment and a sweep gas only contactor. Is the latter experiment presented in one of the measurements explained later? Regarding Fig. 3a, shouldn't this correspond to the case +Gas-Heat (which I do not see)?

2. I see from Fig. 3b that shifting from the heating pad to the lamp brings the flux from 80.2 to 218.5 (in $\text{mmol}/\text{m}^2/\text{min}$). This is the 170% increase mentioned at line 191. Between line 191 and 194 is stated that efficiency could not be compared. I think that at least energy inputs in the two cases should be analyzed. For the light case a 300W lamp is employed, which delivers $1500\text{W}/\text{m}^2$. The electric pad is at 65 degrees. I see the authors implemented a COMSOL model as well. Could they estimate which is the average power per unit area that is transferred from the pad to the channel? Could they compare this power to the light power absorbed by the solution (which should be quite high since it is stated – line 188- that the light is blocked by the nanoparticles in the countercurrent channel)?

3. Assuming the energetic balance proves that light-induced separation outperforms the electric one, I suggest the authors to briefly provide an explanation for it. Is it an increase in volatility of the gas or some other mechanism?

Minor comments are:

- In the SI I cannot see the equation numbers where thermal conductivities are calculated.
- Line 22: after 'including' I would put a colon
- Line 64-65: what does trigger the release in these fishes? I guess it is a chemical reaction. It is a

curiosity of mine but maybe the authors could just briefly mention what it is.
I suggest the publication after the comments have been addressed. In particular the energy balance should be tackled and/or clarified.

Reviewers' comments:

Reviewer #2 (Remarks to the Author):

I thank the authors for their thoughtful responses to my original critiques. These helped improve the article, although there are still some major issues. None of these show-stoppers, and the article should still be considered for eventual publication in Nature Communication.

The short version of my critique is that it appears that the authors do not want to tangibly compare their method to any other method of separating CO₂ from MEA and instead rely on circumstantial discussion to prove their method should be capable of producing purer gas streams than other methods. Indeed, the article would benefit greatly if their new method were to be compared on an energy or area basis. The writing is still difficult to understand throughout the article.

I will go through the responses in detail one by one.

Comment 1. Concerning not discussing countercurrent membrane mediated stripping.

The authors address this well enough in that they at least mention it and explain why their device can offer advantages over other methods circumstantially.

A concern throughout is the fact the authors only seem to benchmark against their own device and not against any kind of standard. I am guessing no such standard exists, but coming up with some comparison metric would substantially help the article.

- 1) We appreciate the suggestion and understand the issue. To clarify, we sought to compare our device to the control experiments, because it was meant to show how the counter current amplification could enhance the activity of a process. This device is not optimized for CO₂ release nor did we want to leave people with that impression. In fact, one of the major challenges to measuring an efficiency is calculating the amount of heat that is being applied to the liquid vs. heating the apparatuses.

Our design does offer improvements in recovering gas and does have unique aspects that we try to highlight. However, we want to make readers aware of both the advantages and limitations. So, we have presented a comparison to existing commercial carbon sequestration systems on line 269.

- a. "In its current form, our device is several orders of magnitude less efficient than existing commercial carbon sequestration methods (80 MWh/tonne CO₂ vs. 0.35 MWh/tonne CO₂). Use of heat exchangers coupling the absorption step with the release stage allows a significant reduction in the energy required for the entire cycle. Future work will focus on integrating heat exchange between absorption and desorption and minimizing the heat loss from the face opposite the heating device. The device, as it is, serves as a proof of concept application of the countercurrent amplification motif seen in fish. Demonstrating how a simple architectural change can lead to an increase in the output of a gas desorption device."

We feel this is a fair comparison and that it highlights our intent without over-selling the technology.

Comment 2. Concerning Wicke-Kallenbach

The authors measure the concentration of the gas when no sweep is used, but do not report flow rates, only concentration. Somehow a 68% pure stream leaving the system is observed when operating with no sweep at 35 °C. What the other 32% of the stream was is never addressed.

- 1) We thank the reviewer for pointing this out and apologize for the confusion. We did not use a sweep gas as the chamber pressurizes itself. We were interested in what the limit of CO₂ purity would be without any sweep. We have clarified this point in line 163
 - a) “The head space of the sensor unit is initially filled with ambient air which accounts for the remainder of the gas composition at the experiment end.”

Comment 3. Authors should have some discussion of where their separations device would operate and what it may replace (absorber-stripper operations)

- 1) We appreciate the suggestion and have included a description of where in the carbon capture stack optimized iterations of our device could be used at line 252. Again, we want to emphasize that our intention here was not to replace any specific part of an exchange process, but to highlight a new exchange process for gas. While such a device might eventually replace a functional unit of an exchanger, this is a demonstration of basic principle and the device is not intended to directly replace any component. However, we do appreciate that it should be clearer how to use this new method and device, so we have included the following sentence.
 - a) “For CO₂, a countercurrent amplifier could be used to lower the energy cost of the stripping process of CO₂ associated with carbon capture operations.”

The authors have addressed my concern by added this discussion at the back end of the paper, where it really should have gone up front to help the reader understand the purpose of the work.

We apologize and this draft seeks to strike a balance between those two options.

Comment 4: OK

Comment 5: See response to Comment 1

Comment 6: OK

Comment 7: OK

Comment 8. How can this device reach similar mass transfer areas to other devices such as hollow fibers?

- 1) This is a great question. This report is about both a process and a device. The device here is crude and modular to allow us to do appropriate controls and to vary parameters quickly. We plan to reach similar mass transfer areas to hollow fibers by using hollow fibers as our exchange device. In current experiments, we use existing hollow fiber membranes to serve the same purpose as the acrylic block in this paper. We have made reference to this continuation on line 233.
 - a. “Our current design allows the device to be easily scaled using common fabrication techniques including laser etching, or CNC machining in addition to pursuing the use of hollow fiber membrane contactors”

As discussed in Comment 1, benchmarking this device against anything else would help the paper substantially. The authors argue their Comsol modelling shows that if they were to shrink down the size of

the channels they can increase purity in the channel. That said, nowhere in the SI (Section 4.3) do they actually show comparisons between the 1mm channel depth and the 0.1 mm channel depth. The only place we see this reported is figure 3d. The authors really should include the plots comparing concentration of the two channel depths in the SI in more detail.

- 1) We have included an overlay of the requisite parts of figures S18 and S19 to better illustrate the change in calculated gas concentration for a given channel depth. Please see Figure S20.

Comment 9: OK

Comment 10: OK

Comment 11: The writing has been improved, but is still confusing to the reader. Perhaps having an unfamiliar but technically savvy colleague go through the article will help clarify this article.

New comments:

Figures 2e and 4b are never directly referenced or explained in the manuscript, as far as I can tell. Why is the flux dropping off strongly near the end of the experiment? Did the heat/light turn off, and if so, why isn't that noted in the plot.

- 1) First, we apologize for this oversight. We did not sufficiently explain the experimental design and have worked to amend this both in the text and the figure. Yes, the lamp was turned off once the gas elution reached an equilibrium.
- 2) We have included references to the above mentioned figures in the text. We have also updated the flux plots to make it clear that the lamp/heating device was shut off towards the end of the experiment.
 - a. Line 123 – “Under these conditions, CO₂ was released at a rate of 6.4 μl/min (80.2 mmol·m⁻²·min⁻¹) with an internal temperature of ≈ 40 °C (Fig. 3a). See Figure 2e for an example of the experimental output.”
 - b. Line 213 – “The release rate of O₂ was 0.8 ml/min (10.2 mmol·m⁻²·min⁻¹) for the PFO/O₂ system – 8 times less than the comparable experiment with the MEA/CO₂ system (Fig. 4). See Figure 4b for an example of the experimental output.”

Incorrect units:

Line 288 The gas detection system was designed to pick up gas flow rates in the 0.1 – 50 μm range. Gas flow rates, length units? Line 208, units of “rate of O₂ production was 0.8 ml/min” I am almost positive based on context this should be μL/min

Thanks for pointing these out, we have fixed these errors with the correct units.

SI:

The calculation of thermal resistance (line 57 of the SI) seems to have a typo. Their solution is right, but the value 0.000020 should be 0.000040.

We appreciate this error being brought to light. It has been corrected.

Figure S9. The figures are not labelled A and B, although the captions calls on them as though they are. Also the y-axis on figure S9 Right (B?) likely should read Ar Flux (units) since there is no O₂ in that experiment?

- 1) We have reworded the callouts to remove any confusion about which figure is which. The units are O₂ flux. We have included a more detailed explanation of the purpose of the experiments in line with the figures.
 - a. Line 94 - "To verify that the measured oxygen was not simply from atmospheric oxygen diffusing through the tubing used to transfer capture or sweep gas to the device, we bubbled either oxygen or argon in the perfluorooctane reservoir. Any oxygen seen when argon was bubbled should be the result of atmospheric leakage into the system."

Reviewer #3 (Remarks to the Author):

The manuscript has improved and it is now well written. However, there are some points that should be addressed. The main concern is about the energy comparison between the different heating methods. Main comments are:

1. The 242% increase is mentioned to be the improvement between the light induced experiment and a sweep gas only contactor. Is the latter experiment presented in one of the measurements explained later? Regarding Fig. 3a, shouldn't this correspond to the case +Gas-Heat (which I do not see)?

- 1) We have changed the wording of this claim to avoid any misunderstanding. We regret any confusion that it may have caused. The 240 % improvement is from the (- Gas, - Heat) to the (+ Gas, + Heat) system. The text at line 56 hopefully makes this clear now. The reviewer makes a good point the best efficiency comparison would be with the +Gas-Heat system. However, this did not yield reliable, measurements of CO₂ as the amount was too low. As such, we could not make that direct comparison. We amended the text for clarity.
 - a. "We observed a 240 % increase in gas release rate when comparing a device where gas and heat is allowed to diffuse between adjacent capture fluid channels compared to a device where no diffusion could take place."

2. I see from Fig. 3b that shifting from the heating pad to the lamp brings the flux from 80.2 to 218.5 (in mmol/m²/min). This is the 170% increase mentioned at line 191. Between line 191 and 194 is stated that efficiency could not be compared. I think that at least energy inputs in the two cases should be analyzed. For the light case a 300W lamp is employed, which delivers 1500W/m². The electric pad is at 65 degrees. I see the authors implemented a COMSOL model as well. Could they estimate which is the average power per unit area that is transferred from the pad to the channel? Could they compare this power to the light power absorbed by the solution (which should be quite high since it is stated – line 188- that the light is blocked by the nanoparticles in the countercurrent channel)?

- 1) We thank the reviewer for this suggestion. Details on the derivation can be found in the supporting information section 4.4
 - a. Line 198 - "Calculating the efficiency of the system based solely on the energy entering channel system, either calculated with COMSOL in the case of the heating pad, or estimated from the surface area of the channels and the intensity of the lamp in the case of the photothermal system gives efficiencies of 1500 MWh/tonne CO₂ and X MWh/tonne CO₂ respectively"
 - b. Line 268 – "In its current form, our device is several orders of magnitude less efficient than existing commercial carbon sequestration methods (80 MWh/tonne CO₂ vs. 0.35 MWh/tonne CO₂)."

3. Assuming the energetic balance proves that light-induced separation outperforms the electric one, I suggest the authors to briefly provide an explanation for it. Is it an increase in volatility of the gas or some other mechanism?

We thank the reviewer for this comment. We have examined the photo-thermal phenomenon in more detail in another set of publications: DOI:[10.1039/C4EE01047G](https://doi.org/10.1039/C4EE01047G). DOI:[10.1021/acsami.5b08151](https://doi.org/10.1021/acsami.5b08151). In these, we describe a series of observations that sum up to the fact that, (1) the process does not need to heat the whole solution to remove gas, (2) the process provides concentrated gas within bubbles, and (3) the actual temperatures within the bubbles are likely not those seen for equilibrium separations.

Minor comments are:

- In the SI I cannot see the equation numbers where thermal conductivities are calculated.
- Line 22: after 'including' I would put a colon

We have addressed these comments.

• Line 64-65: what does trigger the release in these fishes? I guess it is a chemical reaction. It is a curiosity of mine but maybe the authors could just briefly mention what it is.

- 1) We appreciate the reviewer's curiosity. We initially left the information out to avoid going too deep into the biology, but we agree that it's an interesting feature of the natural system and have included brief references to the mechanism at play.
 - a. Line 9 – "At the point where the channel bends back on itself, a triggered change in solubility, due to the interaction of released lactic acid and a pH sensitive hemoglobin, creates a concentration gradient across the two channels."
 - b. Line 35 – "Blood flowing into the swim bladder system passes by a loop adjacent to a gas storage bladder where the release of oxygen from hemoglobin is triggered due to the interaction of released lactic acid and a pH sensitive hemoglobin. Now rich in free oxygen, the blood flows back alongside the incoming channel where free oxygen can diffuse into the incoming channel, increasing the concentration of hemoglobin bound oxygen."
 - c. If the reviewer's curiosity is still piqued, we suggest taking a look at this paper by Brittain for a discussion on the structural origin of the effect and a look at the binding affinity curves - <http://www.ncbi.nlm.nih.gov/pubmed/15598496>

I suggest the publication after the comments have been addressed. In particular, the energy balance should be tackled and/or clarified.

We want to take the time to thank the reviewers for all their hard work and insight as well as you for helping us improve the paper.

REVIEWERS' COMMENTS:

Reviewer #2 (Remarks to the Author):

I thank the authors for patiently and carefully going through this (unfortunately rather lengthy) review process. This paper should be accepted for publication.

Reviewer #3 (Remarks to the Author):

The work has improved. It can now be considered for eventual publication after minor revisions. See following comments:

My comments are:

1. Photothermal heating: paragraph starting at Line 98 and/or Line 179: the authors place carbon black nanoparticles in the fluid within the releasing outgoing channel to enhance the system performance. As I understand, still the particles are spread within the volume of the flow. If it is so, recently, Dongare et al.,(see reference below) found that by applying carbon black nanoparticles directly on the surface of a membrane (which in their work is used for membrane desalination) allowed light to heat conversion to be localized exactly where the diffusive species starts penetrating the membrane. I wonder if the same concept could be applied here. The efficiency may increase especially for thicker channels where the gas can be otherwise released relatively far from the semi-permeable membrane. The authors may want to comment on this.

2. COMSOL model: simulations have been now used to estimate the energy efficiency of the system. Did the author also calculate the actual release rate in the different cases? From the SI it seems they have already implemented different models. In Fig. 4b they show some dotted curves which seem calculations, are they? In any case I would recommend adding more details in the caption to describe the curves. As of now, the figure is not clear.

3. Abstract, Line 10-11: I would move "We have developed an artificial countercurrent amplification system for gas concentration." before Line 16: "Our system...".

4. Line 280: after "fish" I would remove the dot and insert "by".

References

REFERENCE:

Dongare, P. D.; Alabastri, A.; Pedersen, S.; Zodrow, K. R.; Hogan, N. J.; Neumann, O.; Wu, J.; Wang, T.; Deshmukh, A.; Elimelech, M.; Li, Q.; Nordlander, P.; Halas, N. J., Nanophotonics-enabled solar membrane distillation for off-grid water purification. *Proceedings of the National Academy of Sciences* 2017, 114 (27), 6936-6941.

Reviewer #2 (Remarks to the Author):

I thank the authors for patiently and carefully going through this (unfortunately rather lengthy) review process. This paper should be accepted for publication.

We appreciate the reviewers comments and suggestions throughout the review process.

Reviewer #3 (Remarks to the Author):

The work has improved. It can now be considered for eventual publication after minor revisions. See following comments:

My comments are:

1. Photothermal heating: paragraph starting at Line 98 and/or Line 179: the authors place carbon black nanoparticles in the fluid within the releasing outgoing channel to enhance the system performance. As I understand, still the particles are spread within the volume of the flow. If it is so, recently, Dongare et al.,(see reference below) found that by applying carbon black nanoparticles directly on the surface of a membrane (which in their work is used for membrane desalination) allowed light to heat conversion to be localized exactly where the diffusive species starts penetrating the membrane. I wonder if the same concept could be applied here. The efficiency may increase especially for thicker channels where the gas can be otherwise released relatively far from the semi-permeable membrane. The authors may want to comment on this.

1. We appreciate the suggestion. We have mentioned the work and its potential benefit in line 252 of manuscript where we discuss possible methods to increase efficiency.
 - a. "Shifting towards the use of higher efficiency LED lights in addition to work by Halas et al. demonstrating increased efficiency in cross membrane transport using nanoparticles isolated to the surface of the membrane present clear avenues towards optimizing the system.³⁷"

2. COMSOL model: simulations have been now used to estimate the energy efficiency of the system. Did the author also calculate the actual release rate in the different cases? From the SI it seems they have already implemented different models. In Fig. 4b they show some dotted curves which seem calculations, are they? In any case I would recommend adding more details in the caption to describe the curves. As of now, the figure is not clear.

1. We originally were using the full geometry to try and optimize the system, but the computation time with the reaction kinetics included made troubleshooting the model difficult compared to the simplified geometry. We believe the simplified geometry still provides valuable information on the relative changes in the system when altering channel size, membrane thickness, or flow rates.
2. We have also clarified the caption for figure 4b. The multiple traces (solid, dashed, dots) represent different experimental runs.

- a. “Figure 4: (a) Comparison of oxygen flow with oxygen or argon bubbled into the capture fluid reservoir. (b) Representative oxygen data output from the countercurrent amplifier showing three separate trials (Blue O2 flux, Red temperature)”

3. Abstract, Line 10-11: I would move “We have developed an artificial countercurrent amplification system for gas concentration.” before Line 16: “Our system...”.

We agree this makes the paragraph clearer. The changes have been adopted.

4. Line 280: after “fish” I would remove the dot and insert “by”.

We have included this suggestion in the revised manuscript.

We have also reformatted the work to fit the editor’s recommendation for the journal. You will find that some paragraphs have moved from the introduction to the results section.

References

REFERENCE:

Dongare, P. D.; Alabastri, A.; Pedersen, S.; Zodrow, K. R.; Hogan, N. J.; Neumann, O.; Wu, J.; Wang, T.; Deshmukh, A.; Elimelech, M.; Li, Q.; Nordlander, P.; Halas, N. J., Nanophotonics-enabled solar membrane distillation for off-grid water purification. Proceedings of the National Academy of Sciences 2017, 114 (27), 6936-6941.